# Fully Automatic Landmarking of Syndromic 3D Facial Surface Scans Using 2D Images

**DOI:** 10.3390/s20113171

**Published:** 2020-06-03

**Authors:** Jordan J. Bannister, Sebastian R. Crites, J. David Aponte, David C. Katz, Matthias Wilms, Ophir D. Klein, Francois P. J. Bernier, Richard A. Spritz, Benedikt Hallgrímsson, Nils D. Forkert

**Affiliations:** 1Biomedical Engineering Graduate Program, University of Calgary, Alberta, AB T2N 4N1, Canada; 2Department of Radiology, Alberta Children’s Hospital Research Institute and Hotchkiss Brain Institute, Cumming School of Medicine, University of Calgary, Alberta, AB T2N 4N1, Canada; srcrites@ucalgary.ca (S.R.C.); matthias.wilms@ucalgary.ca (M.W.); nils.forkert@ucalgary.ca (N.D.F.); 3Department of Cell Biology and Anatomy, Alberta Children’s Hospital Research Institute and McCaig Bone and Joint Institute, Cumming School of Medicine, University of Calgary, Alberta, AB T2N 4N1, Canada; jose.aponte@ucalgary.ca (J.D.A.); david.katz@ucalgary.ca (D.C.K.); bhallgri@ucalgary.ca (B.H.); 4Program in Craniofacial Biology and Department of Orofacial Sciences, University of California, San Francisco, CA 94143, USA; Ophir.Klein@ucsf.edu; 5Department of Medical Genetics, Alberta Children’s Hospital Research Institute, Cumming School of Medicine, University of Calgary, Alberta, AB T2N 4N1, Canada; fpbernie@ucalgary.ca; 6Human Medical Genetics and Genomics Program and Department of Pediatrics, University of Colorado School of Medicine, Aurora, CO 80045, USA; Richard.Spritz@ucdenver.edu

**Keywords:** facial landmarking, genetic syndrome, 3D surface scan

## Abstract

3D facial landmarks are known to be diagnostically relevant biometrics for many genetic syndromes. The objective of this study was to extend a state-of-the-art image-based 2D facial landmarking algorithm for the challenging task of 3D landmark identification on subjects with genetic syndromes, who often have moderate to severe facial dysmorphia. The automatic 3D facial landmarking algorithm presented here uses 2D image-based facial detection and landmarking models to identify 12 landmarks on 3D facial surface scans. The landmarking algorithm was evaluated using a test set of 444 facial scans with ground truth landmarks identified by two different human observers. Three hundred and sixty nine of the subjects in the test set had a genetic syndrome that is associated with facial dysmorphology. For comparison purposes, the manual landmarks were also used to initialize a non-linear surface-based registration of a non-syndromic atlas to each subject scan. Compared to the average intra- and inter-observer landmark distances of 1.1 mm and 1.5 mm respectively, the average distance between the manual landmark positions and those produced by the automatic image-based landmarking algorithm was 2.5 mm. The average error of the registration-based approach was 3.1 mm. Comparing the distributions of Procrustes distances from the mean for each landmarking approach showed that the surface registration algorithm produces a systemic bias towards the atlas shape. In summary, the image-based automatic landmarking approach performed well on this challenging test set, outperforming a semi-automatic surface registration approach, and producing landmark errors that are comparable to state-of-the-art 3D geometry-based facial landmarking algorithms evaluated on non-syndromic subjects.

## 1. Introduction

Due to the diversity, complexity, and rarity of genetic syndromes, one of the primary difficulties in properly treating afflicted patients is recognizing their condition in the first place. Although exome and genome sequencing have improved diagnostic procedures, genetic tests can be expensive, genetic experts are often scarce, wait times for genetic consultations can be long, and there are still many conditions for which a genetic test is not available. Without a diagnosis, families and patients affected by genetic diseases must sometimes proceed without even basic information regarding health and developmental outcomes, let alone tailored clinical care. Even today, obtaining a definitive diagnosis for a rare genetic disease is often an arduous process that can take years or even decades [1]. Advancements in gene-based technologies are essential to improving diagnostic procedures but complementary strategies ought to be pursued as well. Computer-assisted phenotyping is one strategy that can make use of inexpensive and widely available technologies for rapid syndrome screening.

Dysmorphic facial features (abnormal facial features) occur in over 1500 different human genetic syndromes, and more subtle facial differences are associated with many more [2]. The facial shape associated with a syndrome can be quite distinctive, and previous work has shown that facial shape measurements are useful diagnostic indicators for many genetic syndromes [3,4]. An experienced clinical geneticist will often use the facial gestalt as a preliminary diagnostic test before continuing with genetic testing. Unlike human clinicians, however, who may be able to reference hundreds or even thousands of previous cases if they are exceptionally experienced, computers are capable of referencing millions of facial scans. Additionally, the information extracted by computers from facial scans can easily be shared electronically. The development of robust and fully automatic computational pipelines to analyze facial morphology would allow clinicians all across the globe to utilize a unified quantitative understanding of syndromic facial morphology, in the form of statistical models and associated software, to support clinical decision making processes. Once developed, access to this technology could be provided at very low cost. Syndrome diagnosis tools that use 2D color facial images have been developed [5]. However, the use of 3D surface scans has the potential to improve results and provide additional information to clinicians. A generative 3D statistical model of syndromic facial shape would not only provide diagnostic predictions, but would also be able to show the particular facial morphological effects that are associated with each genetic syndrome.

The most widely used method for 3D morphological analysis (geometric morphometric analysis) requires the identification of a set of anatomically homologous points (landmarks) across all subjects to be measured and analyzed. The landmark set typically consists of a set of distinguishable anatomical features (e.g., the tip of the nose, or the corners of the mouth) distributed across the structure of interest in such a way as to capture the overall morphology. By comparing the relative locations of these landmarks, a Euclidean metric space describing shape can be defined, and population statistics within the shape space can be calculated [6,7]. Unfortunately, the identification of landmarks is often the most difficult, time consuming, and error prone stage in shape analysis pipelines. Although manual identification of landmarks is robust, the approach becomes highly impractical when the subjects that need to be analyzed number in the hundreds or thousands, or when the scans need to be processed in near real time. Additionally, inter- and intra-observer bias in manual landmark placements can be difficult to identify and correct for during data analysis. For these reasons, robust automated 3D facial measurement tools are highly desirable.

### Related Work

A number of automatic and semi-automatic approaches for sparse 3D facial landmarking have been previously developed that use geometric information. A thorough review of recent approaches is provided by Manal et al. [8]. Briefly described, some of these methods are based primarily on geometric heuristics [9,10], while others utilize geometrically defined point-wise shape descriptors in combination with statistical or machine learning models to identify landmarks [11,12]. The mean error of these approaches ranges from 1.3 to 5.5 mm. However, it should be noted that different algorithms often identify different landmarks and are evaluated on different datasets so that the results of different experimental setups cannot be directly compared. Additionally, due to the unstructured nature of 3D surface scan data, estimating the pose of a subject’s face in a noisy scan can be difficult, which is an important limitation because many geometric landmarking approaches assume a certain initial orientation of the face.

Approaches to facial shape measurement have also been extended from sparse landmarks to dense surfaces [13]. By treating the vertices of an atlas polygonal mesh as anatomically homologous landmark points, tens of thousands of points can be used to characterize facial shape. These dense shape measurements are capable of capturing subtle information about facial shape that is not represented in the relative locations of a few sparse landmarks. Additionally, a variety of automatic methods to register atlas surface meshes to subject scans have been developed. For example, Amberg et al., extended the iterative closest point algorithm to enable non-rigid deformations of the atlas mesh and regularize the deformations using the assumption that the deformations are locally affine [14,15]. The MeshMonk software package for 3D facial shape measurement uses a variant of this non-rigid iterative closest point algorithm [16]. Vetter et al., have also developed an approach that allows integrating statistical priors defined as Gaussian processes to constrain and regularize the deformations of the atlas mesh [17]. Unfortunately, algorithms that rely on iterative closest point correspondence estimation are known to be sensitive to initialization. They typically require manually identified guide points to coarsely align the surface model to the subject scan before registration or to guide the registration algorithm directly. If such points are not provided, the registration may converge to a non-optimal solution. Thus, the need for robust identification of guide points, that is, sparse landmarks, is still present for dense shape measurement approaches.

One approach, which diverges from the previously described approaches, is to use projection transformations to render 2D images of the 3D facial meshes. This transforms the problem of identifying facial landmarks on a 3D surface scan into the problem of identifying 2D facial landmarks on an image. Once identified, the 2D landmarks can be projected back onto the 3D mesh using a ray casting algorithm. Booth et al. [18] employed this approach in order to construct a large scale 3D facial shape model but did not compare the 3D landmarks produced by the algorithm to manually identified landmarks. In the domain of 3D human facial analysis, transforming data from 3D scans to 2D images using a projection transformation is exceptionally useful as it allows for the use of extensively studied 2D facial landmarking algorithms [19] and large databases of labeled facial images. Image-based deep learning models, similar to the ones employed in this work, have seen wide and successful application to other biometric tasks [20]. The performance characteristics of 2D image-based approaches applied to 3D surface scans have not been well studied however, and it is not clear whether 2D facial recognition and landmarking models trained on large databases of normative facial images will generalize well to syndromic subjects with moderate to severe facial dysmorphia.

The main aim of this work was to extend a state-of-the-art image-based 2D facial landmarking algorithm for the challenging task of 3D landmark identification on subjects with genetic syndromes, many of whom have moderate to severe facial dysmorphia. Due to the rarity of genetic syndromes, syndromic facial phenotypes are typically absent from the datasets used to train and evaluate facial landmarking algorithms, and to develop facial shape atlases. The performance of the automatic algorithm is compared to both inter- and intra-observer errors on the same test set as well as to a semi-automatic registration-based approach. The feasibility of using this algorithm in a computer-aided syndrome diagnosis application is also discussed.

## 2. Materials and Methods

### 2.1. Data Description

The 444 3D facial scans used in the analysis presented here were acquired using a 3DMD facial imaging system (www.3dmd.com) and collected by members of the FaceBase Consortium (www.facebase.org). The subjects were recruited through clinical geneticists at different sites across North America. The subjects ranged in age from under 1 year to 75 years, and 369 subjects had a genetic syndrome that is known to be associated with facial dysmorphology. The age and syndrome distributions for the test set are shown in Figure 1 and Figure 2. Each facial scan was landmarked by two human observers with a set of twelve landmarks (Figure 3). A subset of 100 scans was also landmarked a second time by the same observer in order to estimate intra-observer agreement. The second landmarking session took place one day after the first was completed and the scans were presented to the observer in a different order than in the first session. Eight of the landmarks are of type 1, meaning that they are identifiable using histological information. The other four landmarks are of type 2, meaning that they are identifiable using geometric information only. The landmark set was selected such that the landmarks captured the position and orientation of a face as well as some coarse morphological information about the central facial substructures (eyes, nose, mouth). The test set of scans was specifically selected to represent the distribution of subjects that might be encountered at a clinical geneticists office. This study was conducted in accordance with the Declaration of Helsinki and ethics approval has been granted by the Conjoint Health Research Ethics Board (CHREB) at the University of Calgary. The 3D scans used in this work are freely available upon reasonable request to the FaceBase Consortium (www.facebase.org). The code is publicly available on github (github.com/JJBannister/3DFacialMeasurement).

### 2.2. Image-Based Landmarking Algorithm

The automatic image-based 3D landmarking algorithm evaluated in this work takes a polygonal surface mesh containing a human face as input and returns a set of 3D facial landmarks. This algorithm extends a 2D image-based facial landmarking model [21] to identify 3D landmarks on 3D surface scans. A broad overview of the 3D image-based landmarking algorithm will be provided first, followed by details of the various components.

#### 2.2.1. Overview

The algorithm begins by determining the coarse location of the face in the polygonal mesh. Therefore, an array of virtual cameras is positioned around the mesh to capture 2D renderings from different view points. A 2D facial detection model is applied to each rendered image until a face is located in one of the images. Next, the algorithm identifies a frontal camera position. Therefore, a 2D facial landmarking model is used to locate a set of 2D facial landmarks in the previously selected image and the 2D landmarks are projected back onto the 3D mesh using a ray-casting algorithm. A new virtual camera position is then computed using the newly identified 3D landmarks, which centers the face in the camera’s field of view. The frontal camera position is refined once before identifying the final landmark positions by re-identifying a set of 3D landmarks using the frontal camera rendering and re-computing the frontal camera position. Lastly, the final 2D landmark positions are identified using the refined frontal image and projected back onto the 3D mesh. Figure 3 and Figure 4 show various stages of the algorithm when applied to an example 3D surface scan.

There are four main configurable components to the algorithm just described: the initial camera array used to coarsely locate the face, the algorithm for computing the frontal camera position from a set of 3D landmarks, the 2D image-based facial detection model, and the 2D landmarking model.

#### 2.2.2. Component Details and Specification

In this work, the facial detection model is a deep convolutional regression network trained to estimate bounding boxes around human faces in images. The input to the model is a 2D image and the output is a variable number of bounding boxes and associated confidence scores. The assumption was made that no more than a single face was present in each scan. Thus, the bounding box with the highest confidence score was selected in the case that the model outputs more than one bounding box. The facial landmarking model used in this work was an ensemble of regression trees as described by Kazemi et al. [21]. The model consists of a cascade of regressors trained using the gradient boosting tree algorithm. The input to the model is a 2D image and the output is a set of 68 2D facial landmark points. Both models were trained on large databases of normative facial images and represent state-of-the-art techniques for face detection and landmark localization in 2D images. Both trained models are available through the Dlib library (github.com/davisking/dlib-models).

To configure the other components for this particular application, prior information about the nature of the test data was used. Additionally, a single subject scan (not included in the test set) was used for qualitative experimentation to help select robust parameters.

For the definition of the initial camera array used to coarsely locate faces in a polygonal mesh, the algorithm makes use of two pieces of prior information about the 3D scanning system used to capture the 3D scans. The reference frame of the 3D scanner was defined such that the scanner pointed in the −z direction and the *y* direction pointed vertically up. Accordingly, for a given mesh, we defined an initial array of five cameras. Each camera’s focal point was defined by the centroid of the input mesh, all cameras were oriented such that the *y* direction pointed up in the camera’s reference frame, and the view angle was set to 50 degrees. The central camera was placed 60 cm in the *z* direction from the centroid of the mesh. The four other camera positions were determined by translating the central camera 60 cm in the x,y,−x, and −y directions. Effectively, this provides five different viewpoints of each mesh with wide fields of view such that if there is an identifiable facial structure in the mesh, it will be clearly visible to at least one of the cameras (Figure 4). If no information were available about the input 3D mesh and its orientation, a different distribution and number of cameras could be used (e.g., one that uniformly samples a 3D sphere around the centroid of the mesh). Theoretically, this approach could be used to locate multiple facial structures in complex 3D scenes that contain much more content than just a facial surface and parts of the upper body.

For the determination of a frontal camera position given a set of 3D facial landmarks, a simple geometric algorithm was designed. In this work, we utilized seven 3D landmarks to compute the frontal camera position: ch_l, ch_r, ex_l, ex_r, n, prn, and ls (Figure 3). The focal point of the frontal camera was defined by the prn landmark and the view angle was set to 20 degrees. The location of the camera was set to be 80 cm from the prn landmark along the direction defined by the vector cross product (ch_r−ex_l)×(ch_l−ex_r). The roll of the camera was set to align the vertical axis of the camera with the vector (n−ls) so that faces are oriented vertically within the frontal rendering even if they were tilted during the scan acquisition. The process of identifying landmarks and computing a frontal camera position is repeated twice before identifying a final set of landmarks in order to ensure the landmarks used in computing the frontal camera position are robust.

All rendering and ray casting operations were implemented using the Visualization Toolkit (VTK) library in python. The Dlib models were accessed through the face-recognition package available through the Pypi repository. For a single subject scan, the final implementation of the image-based 3D landmarking algorithm runs in under one second.

### 2.3. Dense Registration Algorithm

As a comparison method for the proposed image-based automatic 3D landmarking algorithm, we also applied the Non-rigid Iterative Closest Point (NICP) registration algorithm [15] to register an atlas facial mesh to the subject scans. The registration algorithm attempts to deform the atlas mesh to match the morphology of the subject scan in such a way that anatomical homology is preserved. This means that after registering the atlas facial mesh to a subject scan, the point on the atlas mesh corresponding to the tip of the nose, for example, would ideally lie at the tip of the nose of the subject scan. Therefore, the registration algorithm can be evaluated in the same manner as a sparse landmarking algorithm by taking the subset of points on the atlas mesh that correspond to the landmark positions and comparing the position of those atlas points at the termination of the registration algorithm to the positions of manually identified landmarks. For a single subject scan, the NICP algorithm runs in five to ten minutes on a mid-range desktop computer.

To construct an atlas mesh, we utilized an age diverse, normative database of 2273 3D facial scans not included in the study test set. A single scan was randomly selected from the database. The scan was re-meshed using a Poisson reconstruction algorithm [22] to ensure a 2-manifold mesh representation and manually cropped to the facial area captured by a typical scan from the database. The mesh was then decimated by a quadratic edge collapse algorithm [23] to 27,903 points. The number of points was selected to provide a surface sampling density that is approximately the same as the raw 3D scans. The mesh was then registered to each scan from the normative database using the NICP algorithm. 24 manually identified landmarks on each scan were used to guide the registrations. Rigid body Procrustes alignment was then used to remove variation associated with translation and rotation from the registration transformations and the final atlas mesh was computed by averaging the position of each point across all registered facial meshes. The landmark positions of the atlas were similarly calculated as the averaged positions of the landmark points across all deformed meshes. The resulting atlas mesh is shown in Figure 5.

As the NICP algorithm includes an optional guide point term in its loss function for pre-identified point correspondences, it is possible to force the landmarks of two scans to be registered to align perfectly. This is the approach that was used during the atlas construction stage. However, as the landmark positions serve as the ground truth used to evaluate the performance of the algorithm, the atlas and scan landmarks were only used to compute an affine transformation that coarsely aligned the atlas to the subject scans in the test set. The affine transformation was selected to minimize the sum of squared distances between the atlas and subject landmarks. In other words, the NICP algorithm was initialized using manually identified landmarks but, the landmarks were not used as guide points within the NICP algorithm. Therefore, the landmark positions computed using the NICP algorithm on the test set of scans can be used to evaluate the ability of the NICP algorithm to deform the atlas to match the morphology of the subject scan. This approach was selected for comparison to the proposed image-based method because it is commonly used for semi-automatic dense surface measurement of 3D facial scans [16,18].

## 3. Evaluation

To evaluate the performance of the proposed image-based landmarking algorithm and the semi-automatic NICP surface registration algorithm, we compared the locations of the selected facial landmarks identified by the two methods to ground truth landmarks identified by two different human observers. First, to establish a baseline level of performance, the inter-observer and intra-observer agreement was computed using manually identified landmarks. Next, the accuracy of each landmarking method was evaluated by computing the Euclidean distances between the landmark positions identified by each method and the human observers. For each method, the distances were computed for both sets of manually identified landmarks and then averaged.

While the accuracy of the algorithms is certainly of interest, the precision of the algorithms is of equal importance. To compare the precision of the proposed landmarking method, the semi-automatic NICP landmarking method, and the manual ground truth landmarks, we analyzed the Procrustes distance from the mean for each subject. Procrustes distance measures the similarity of two shapes and is a commonly used metric on shape spaces. To compute Procrustes distance from the mean, subject landmarks were scaled to have unit centroid size and aligned to one another using rigid-body transformations employing Procrustes alignment. Procrustes alignment is an iterative procedure. In each iteration, the average landmark positions are computed and each set of subject landmarks is aligned to the average landmarks by applying a rigid-body transformation that minimizes the sum of squared distances between the subject landmark configuration and the average landmark configuration. The alignment procedure terminates once the average landmark configuration converges to within a pre-selected threshold. After alignment, the Procrustes distance, the square root of the sum of squared distances between the average landmark configuration and each subject landmark configuration was computed.

## 4. Results

Figure 6 shows a boxplot comparing the Euclidean distances between landmarks identified using the proposed image-based method and the semi-automatic comparison method using the NICP algorithm. Table 1 records the means and standard deviations of the distributions visualized in Figure 6. The overall average intra-observer landmark distance was slightly lower than the overall average inter-observer distance with values of 1.1 and 1.5 mm, respectively. There were no considerable differences in manual landmarking errors between type one and type two landmarks. The overall average distance between landmarks produced by the proposed automatic image-based landmarking algorithm and the manual landmarks was 2.5 mm. Comparatively, the NICP landmarking error was 3.1 mm.

Figure 7 shows a boxplot comparing the subject averaged landmarking errors for the NICP and image-based landmarking methods separated by the diagnostic status of the subject. The mean error was lower for the image-based algorithm across both syndromic and control subjects. All subjects for whom the image-based landmarking algorithm produced an average error above 5 mm had some variety of genetic syndrome.

Figure 8 shows the distributions of Procrustes distances from the mean for each of the landmarking procedures. The distributions show that, compared to a manual approach, the image-based landmarking algorithm produces landmark configurations that are further from the mean shape, including some extreme outliers. Comparatively, the NICP algorithm produced landmarks that were closer to the mean shape compared to both the manual landmarking approach as well as the image-based landmarking approach.

## 5. Discussion

In general, the proposed automatic image-based landmarking algorithm performed well on a challenging test set. The overall average distance from the manual landmarks was lower than that of landmarks produced by the NICP algorithm initialized with manual landmarks. The observed errors are also within the range of state-of-the-art geometry-based landmarking algorithms (1.3–5.5 mm) that were evaluated on different normative facial scan sets, but also using different landmark sets so that a direct comparison is not possible. However, as a standalone shape measurement process, the results are not as robust as a manual landmarking approach, exhibiting lower accuracy and precision.

The evaluation of the accuracy of manual landmarking led to expected results. Inter-observer errors were slightly larger than the intra-observer errors, and the largest Euclidean distance between landmarks placed by two different observers was 10 mm. There were some landmark positions that tended to to have lower manual errors than others, but the differences were not large. The pronasale landmark (prn) had the lowest average intra- and inter-observer errors while the labiale superius (ls) and subnasale (sn) landmarks had the largest.

The accuracy of the image-based automatic landmarking algorithm was surprisingly good considering the nature of the test set. The algorithm recognized and produced reasonable landmark estimates for many subject faces with moderate and severe dysmorphia. Perhaps the most prominent difference between the manual landmarking approach and the image-based automatic landmarking approach was that the tail of the error distribution was longer for the image-based automatic approach. While no single landmark position had a mean landmarking error of over 3.4 mm, there were a small number of subjects with errors over 10 mm (the largest inter-observer error). Only one subject had an average landmark error over 20 mm. All subjects with average errors over 5 mm had some variety of genetic syndrome. For the image-based automatic algorithm, the eye landmarks (en, ex), the nasion landmark (n), and the gnathion landmark (gn) showed larger errors compared to the other landmarks. The nasion and gnathion landmarks are both type two landmarks, and a certain degree of ambiguity in their location is therefore expected. The performance of the automatic algorithm on the endocanthion landmarks was surprising, as they are prominent type 1 landmarks. The right endocanthion had the highest average error of any landmark position, with a value of 3.4 mm. A visual inspection of the subjects with the largest automatic landmark errors in these positions revealed that closed eyes appear to be a common problem. This is an important consideration for future data collection. The performance of the automatic algorithm on the mouth landmarks (ch, ls, li) as well as the nose landmarks (prn, sn) can be considered comparably good.

The accuracy of the semi-automatic NICP algorithm was worse on average compared to the proposed image-based approach. However, the NICP approach produced fewer extreme outliers. One notable result from the analysis of the NICP landmarking accuracy is the difference in accuracy between bilateral landmarks. For all three bilateral landmark positions (en, ex, ch), the NICP landmarking accuracy was worse for the left landmark compared to the right. This pattern is likely a reflection of facial asymmetry in the test set but could also reflect asymmetry in the atlas facial mesh. In this work, no attempt was made to force the atlas mesh to be bilaterally symmetric. Techniques introduced by Luthi et al. [17] could be employed to force bilateral symmetry in the atlas during the registration process. However, this could also have the effect of increasing measurement error in subjects whose faces are moderately or highly asymmetric.

The analysis of Procrustes distances from the mean revealed similar insights to the accuracy analysis. The mean Procrustes distance from the mean shape was slightly smaller in the manual landmarks compared to the image-based automatic landmarks, with values of 0.008 and 0.011 respectively. The distribution of Procrustes distances from the mean also had a longer tail in the image-based landmark set compared to the manual landmark set. These results indicate that the image-based automatic algorithm is less precise compared to a manual landmarking approach and that the automatic approach has a greater tendency to produce extreme outliers. Comparing the distributions of Procrustes distances from the mean produced by the NICP algorithm with the distribution produced by a manual landmarking approach also revealed an important bias in the NICP algorithm. The mean Procrustes distances from the mean for landmarks produced by the NICP algorithm were lower than that of both the manual landmarks and the landmarks produced by the image-based landmarking algorithm. Additionally, the tail of the distribution was shorter for the NICP landmarks. These results indicate that the landmarks and dense shape measurements produced by the NICP algorithm are biased towards the average facial shape. Given the prior assumptions made by the NICP algorithm, this is expected. At the outset of the algorithm, the morphology of the atlas mesh corresponds to an averaged facial shape deformed only by a globally affine transformation and, subsequently, the algorithm resists deformations of the atlas away from the initial shape according to the geometric regularization used within the NICP algorithm. While some form of regularization is necessary to prevent the atlas from deforming in extreme and implausible ways, this form of regularization results in shape configurations that are closer to the average shape compared to manual or image-based configurations. When used in a typical shape analysis application, this could have the effect of softening or obscuring extreme morphological effects. In the context of a computer-aided syndrome diagnosis application that seeks to recognize and differentiate between unusual and often extreme facial morphologies, an average bias in the shape measurement algorithm would be undesirable. Statistical regularization, such as that proposed by Luthi et al. [17], may ameliorate this bias by accounting for common modes of morphological variation within the registration algorithm.

### 5.1. Limitations

The primary limitation of the proposed image-based landmarking algorithm compared to geometry-based landmarking algorithms is the reliance of the proposed algorithm on facial surface color and reflectance information. Unlike geometry-based algorithms, the proposed image-based algorithm cannot be applied to facial meshes without color information. Most modern 3D surface scanners will capture surface color information in addition to geometric information. However, color information is often stripped from facial scans as part of anonymization processes, so this limitation is prohibitive in some circumstances. Additionally, the use of color information introduces a potential source of variability and error. Changes in subject complexion and illumination could influence landmark placement even if the underlying facial geometry does not change. In practice, this introduces the need for additional quality control procedures during the scan acquisition stage. In the context of a 3D facial shape-based software tool for computer aided diagnosis, subjects should be imaged in a controlled environment with a scanner that supports the collection of surface color information. Ideally, a quick visual check of the scan would also be performed to ensure the acquisition is of acceptable quality. This is especially important when imaging young children or subjects with intellectual disabilities who do not want to sit still for a scan. These processes were followed when acquiring the test data used in this study.

### 5.2. Future Directions

Future work will focus on developing and evaluating a fully automatic 3D facial shape-based software application for computer-aided syndrome diagnosis. Therefore, a dense statistical shape model will be created using the complete FaceBase scan set that explicitly incorporates syndromic facial morphological variation. Finally, a software tool will be developed that allows clinicians to automatically measure the facial shape of their patients, compare the facial shape of their patients to the expected facial shapes associated with different genetic syndromes, and to analyse the likelihood of their patient’s facial shape under different syndromic diagnostic hypotheses.

## 6. Conclusions

In this work, we presented an implementation of a fully automatic 2D image-based 3D facial shape measurement algorithm and evaluated the algorithm on a test set of subjects with a variety of genetic syndromes as well as healthy controls. In general, the image-based automatic landmarking approach performed well on this challenging test set, outperforming a semi-automatic surface registration approach, and producing landmark errors that are comparable to state-of-the-art 3D geometry-based facial landmarking algorithms. Additionally, the algorithm is able to landmark a single scan in under one second on a standard desktop computer, making it suitable for near real time applications.

## Figures and Tables

**Figure 1 sensors-20-03171-f001:**
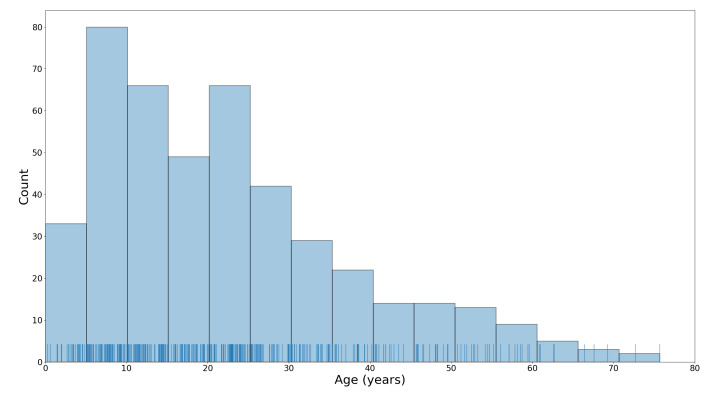
A histogram and rug plot of the test subject ages.

**Figure 2 sensors-20-03171-f002:**
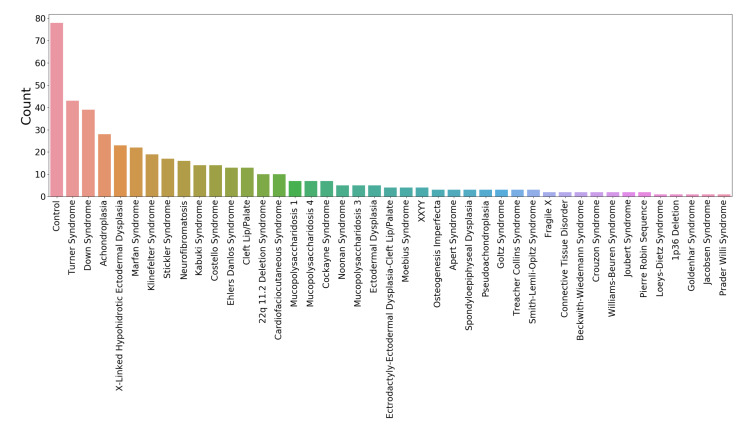
The syndrome distribution of the test set.

**Figure 3 sensors-20-03171-f003:**
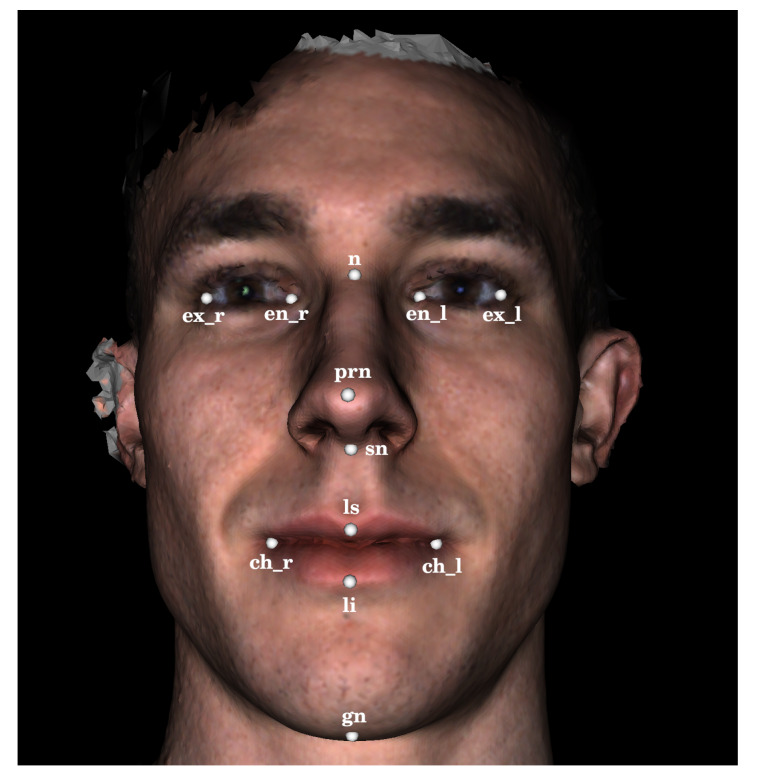
A facial surface scan annotated with the twelve 3D landmarks used in this study. The 3D landmarks were identified by projecting a subset of the 2D landmarks shown in Figure 4 onto the surface scan using a ray casting algorithm.

**Figure 4 sensors-20-03171-f004:**
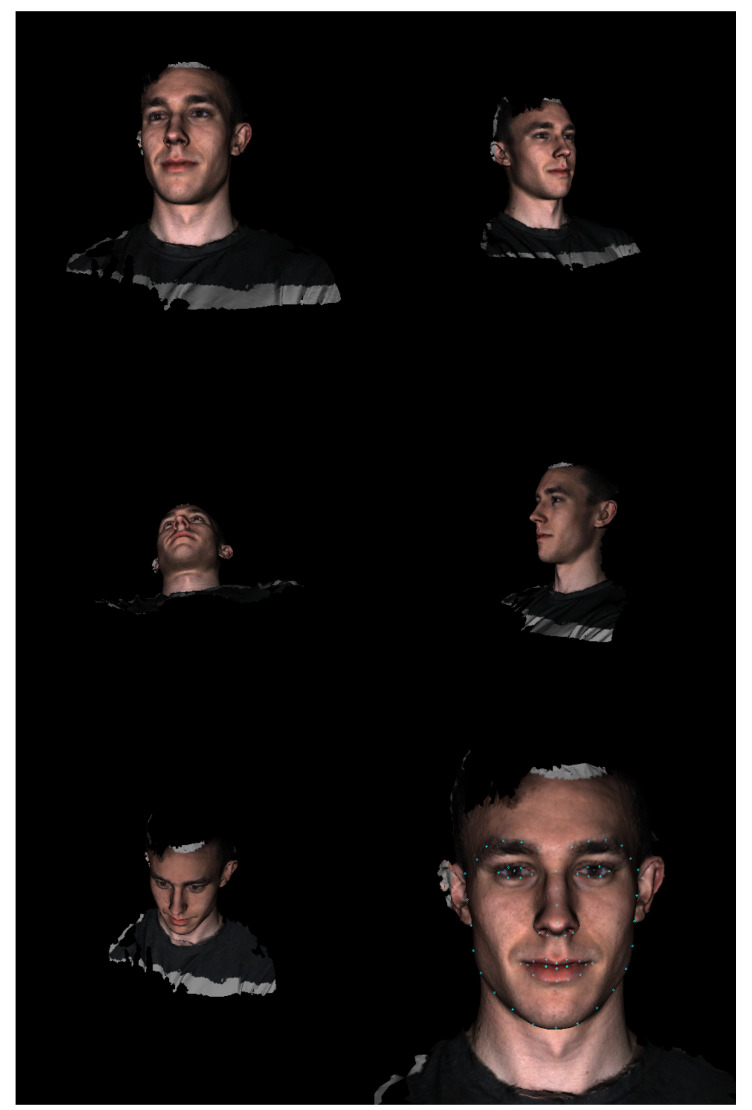
The five initial images of a 3D facial surface scan along with the refined frontal image (bottom right) annotated with the 68 2D landmarks (blue circles) identified by the 2D landmarking model.

**Figure 5 sensors-20-03171-f005:**
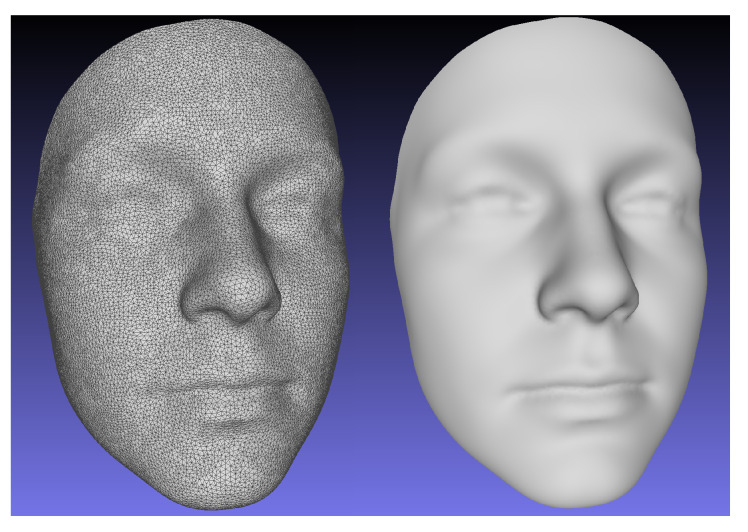
The averaged normative facial mesh used as the atlas for the Non-rigid Iterative Closest Point (NICP) registrations.

**Figure 6 sensors-20-03171-f006:**
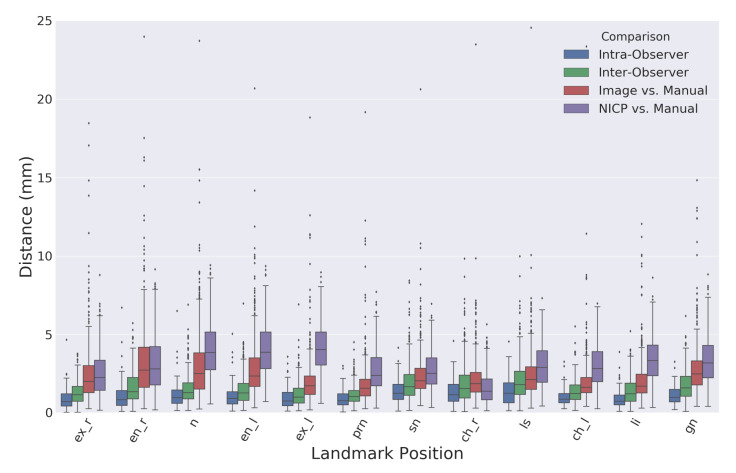
A boxplot comparing the Euclidean distances between landmark positions identified using different landmarking procedures. The boxes extend from the Q1 to Q3 quartile values of the data, with a line at the median. The position of the whiskers is set to 1.5 times the interquartile range (Q3–Q1) from the edges of the box.

**Figure 7 sensors-20-03171-f007:**
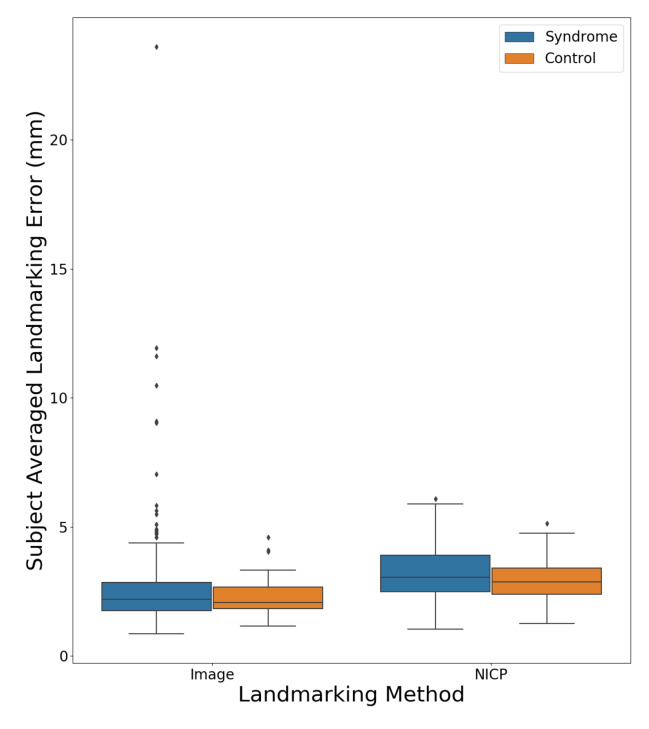
The subject averaged landmark errors for both the NICP and image-based landmarking methods separated by syndrome diagnosis.

**Figure 8 sensors-20-03171-f008:**
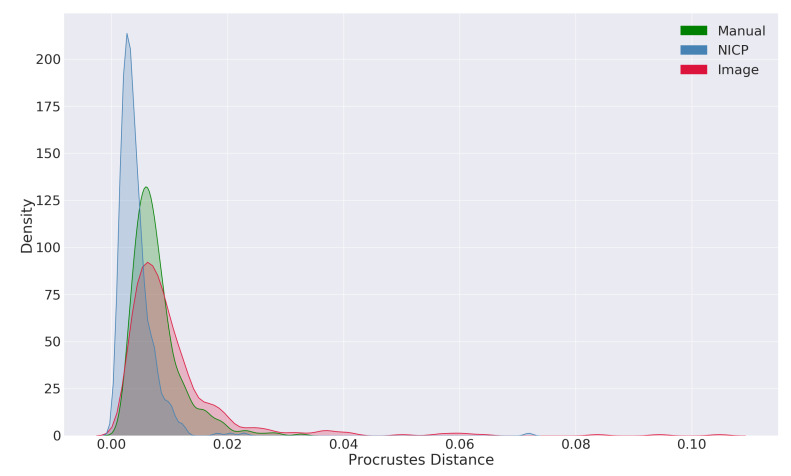
A kernel density plot of Procrustes distances from the mean for different landmarking procedures.

**Table 1 sensors-20-03171-t001:** The mean and standard deviation of the Euclidean distances between different sets of landmarks. All values have units of mm.

Landmark	Intra-Observer	Inter-Observer	Manual vs. Image	Manual vs. NICP
en_r	1.1 (0.7)	1.6 (1.0)	3.4 (2.6)	3.2 (1.9)
en_l	1.1 (0.8)	1.4 (0.9)	2.9 (2.0)	4.0 (1.7)
ex_r	0.9 (0.7)	1.2 (0.7)	2.5 (2.0)	2.5 (1.4)
ex_l	0.9 (0.7)	1.2 (0.8)	2.0 (1.7)	4.1 (1.6)
n	1.2 (0.9)	1.5 (0.9)	3.1 (2.4)	4.0 (1.7)
prn	0.9 (0.5)	1.1 (0.6)	1.9 (1.6)	2.7 (1.3)
sn	1.4 (0.8)	1.9 (1.2)	2.4 (1.6)	2.7 (1.1)
gn	1.1 (0.6)	1.8 (1.0)	2.9 (2.3)	3.3 (1.5)
ch_r	1.3 (0.8)	1.9 (1.3)	2.2 (1.6)	1.6 (1.0)
ch_l	1.0 (0.5)	1.4 (0.7)	2.0 (1.6)	3.0 (1.3)
ls	1.4 (0.9)	2.0 (1.3)	2.5 (1.7)	3.0 (1.3)
li	0.9 (0.6)	1.4 (0.8)	2.1 (2.1)	3.5 (1.5)
Overall	1.1 (0.8)	1.5 (1.0)	2.5 (2.0)	3.1 (1.6)

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
