# Peer review of "Fully Automatic Landmarking of Syndromic 3D Facial Surface Scans Using 2D Images"

_sensors, 2020, doi:10.3390/s20113171_

Round 1
Reviewer 1 Report
The presents a fully automatic state-of-the-art image-based 3D facial landmarking algorithm for subjects with genetic syndromes, who often have moderate or severe facial dysmorphia. Their automatic 3D facial landmarking uses 2D image-based facial detection and landmarking models to identify 12 landmarks on 3D facial surface scans. They evaluated the algorithm on a test set of subjects with a variety of genetic syndromes as well as healthy controls. The goal of the paper is interesting and fully intersects the topic of the Journal. I have only some major comments that I summarize below:
- the authors should better describe the main contribution of the work with respect to the existing literature. The authors should better clarify that the main contribution is the application of state-of-the-art image-based landmarking algorithm to a challenging scenario (test set of syndromic subjects)
- Section II should better clarify the baseline algorithm and any modification/adaptation the authors performed for adapting the algorithm in this challenging scenario. I would suggest that the authors should reorganize this section accordingly.
- I suggest that the authors should perform a statistical comparison (comparing the Euclidean distances between landmark positions identified using different landmarking procedures) for evaluating the performance of the employed algorithm
- How do the authors aim to overcome the limitations described in Section 5.1? How do these limitations may influence the development of a fully automatic 3D facial shape-based software application for computer-aided syndrome diagnosis? Please provide more details about this point.
Reviewer 2 Report
The authors propose an algorithm for landmarking syndromic 3D
facial surface scans using 2D images. There are some major points that need to be addressed:
1. The authors only validate their algorithm on one test set. I think it is not enough. If the authors would like to convince the efficiency of their method, it is more important to validate on different datasets.
2. Fig. 4 was mentioned before Fig. 3 in the text appearance, it is better to switch.
3. The methodology has been explained not clear and it affects the readability. For example, the authors used CNN but did not mention how to construct and how it fitted into the system. What are the input and output of the algorithms?
4. The authors should compare the use of different distance functions (i.e., Euclidean, Procrustes, …) to convince that their choice of distance is the optimal one.
5. References are weak, the authors should add more related works, especially in their methodology. For example, CNN is common and has been used in a lot of works such as PMID: 28643394.
6. Which statistical tests that the authors performed to plot the box chart?
7. The authors should compare their performance results with different published methods and also baseline models on the same dataset.
8. If possible, the source codes should be released for reproducing the results.
9. There are few typos i.e. line 228: "the the accuracy", … The authors should re-check carefully.
Reviewer 3 Report
Review
The main aim of the manuscript is to introduce the development and evaluation of a state-of-the-art technology-based facial landmarking algorithm for the diagnosis of subjects with genetic syndromes, with moderate or severe facial dysmorphia.
The automatic 3D facial landmarking algorithm presented in the manuscript uses 2D image-based facial detection and landmarking models to identify 12 landmarks on 3D5 facial surface scans.
The authors presented a fully automatic 2D image-based 3D facial shape measurement algorithm, and the measurement, evaluation of the mentioned algorithm.
Originality/Novelty:
The main goal of the introduced research is well defined, well implemented in the manuscript.
The results are important in the field of biometrics and face recognition based diagnostics,
mainly related to the early diagnosis of genetic syndromes. The results provide an advance in current knowledge.
Significance: The achieved results are significant and appropriately interpreted in the manuscript.
The authors' conclusions are justified and supported by the results.
Quality of Presentation:
The article was written excellently. The number and relevance of the used references are adequate.
Scientific Soundness: This article has great scientific merit.
The research and the development process were designed correctly, the attached source code is clear, precise work.
The method is described with sufficient details.
English Level:
The English language of the article is appropriate and easy-to-read, understandable.
Reviewer 4 Report
The wprk subject is interesting and you have put good efforts to make it clear and useful to readers.
However, a clear comparison with other state-of-the-art algorithms will be well seen by readers to make the work comparable with the known methodologies and algorithms.
The language is readable, some words are missused like bio-metrics and while:
- bio-metrics (appears in the abstract) should be biometrics, it is not a compound noun;
- while is missused 5 times. Change while into whilst wherever it appears in the text. In all casees it was used to mean 'whilst' not while.
Round 2
Reviewer 2 Report
I appreciate the authors for their efforts on addressing my previous comments. However, there are some major points that have not yet addressed well, or address them not satisfactorily. For examples:
It is important to validate the performance of their models. The authors mentioned that they can't find a similar public dataset for validation, however, in this case, the authors could manually retrieve a new set for validation. Or the authors could use the latest data for validation.
The authors showed source codes in the revised version, however their GitHub link does not contain any instruction for reviewer and users. It makes very difficult for reproducing their results.
The methodology is still not explained well. The authors answered this comment but it seems not satisfactory for me.
As mentioned, Procrustes metric was selected due to it’s applicability in this scenario and because of it’s wide usage within geometric morphometrics. It is not enough evidence to say that if it performed well in the previous works, it can also be good in your data. That's why I suggested to have a comparison in the previous comments.
Which types of errors that the authors used to present the error bars in Fig. 6?
It is necessary to have a comparison to previous methods and previous works.
Round 3
Reviewer 2 Report
My previous comments have been addressed.